# Identifying missed opportunities in tuberculosis preventive treatment care cascade: Analysis of programme data from Maharashtra, India

Anuj Mundra[1*], Tarun Bhatnagar[2☯], Mrinalini Das[3☯], Sandeep Sangale[4], Hemant Patil[5], Abhishek Raut[1], Aniruddha Kadu[6], Rameshwar J. Paradkar[7], Subodh S. Gupta[1], Bishan S. Garg[1]

1 Department of Community Medicine, Mahatma Gandhi Institute of Medical Sciences, Wardha, Maharashtra, India, 2 ICMR-National Institute of Epidemiology, Chennai, India, 3 FIND, New Delhi, India, 4 Joint Director (TB and Leprosy), Maharashtra, India, 5 District TB Officer, Wardha, Maharashtra, India, 6 Consultant, WHO-NTEP technical support network, Maharashtra, India, 7 Assistant Director of Health Services (Leprosy), Wardha, Maharashtra, India

☯ These authors contributed equally to this work.
* anuj.mundra87@gmail.com

## Abstract

Tuberculosis infection is a condition when a person harbours the bacilli without having signs of active disease. In India, over 50% of household contacts of people with pulmonary tuberculosis have the infection. The national tuberculosis elimination programme recommends preventive treatment to such household contacts after ruling out active disease. Maharashtra is one of the bigger states in India with high tuberculosis burden. We analysed the programme data from Maharashtra to describe the tuberculosis preventive treatment care cascade for household contacts of all notified people with pulmonary tuberculosis for the year 2023. Contact tracing was done for 84% of the 133,167 notified people with pulmonary tuberculosis. A total of 406,291 household contacts were enlisted out of whom 386,224 (95%) were screened for symptoms of tuberculosis. 185,502 (45%) household contacts were listed as eligible for tuberculosis preventive treatment, of whom 101,325 (55%) were initiated on tuberculosis preventive treatment. While 41,480 (41%) of those initiated on treatment successfully completed it, treatment outcomes were not recorded for around 57,191 (56%) of them. Tuberculosis preventive treatment completion as well as recording of treatment outcomes was lesser for 6H regimen, among contacts of those seeking care from private sector and people with clinically diagnosed tuberculosis. There were considerable losses at all steps of the TPT cascade. Reasons for losses from the cascade need to be identified and addressed. Strengthening data capturing and reporting mechanisms and developing decentralized mechanisms for identification, evaluation and TPT provision are necessary for improvement in service delivery and utilization. Aligning the national TB report with the guidelines by including household contacts of all notified persons with pulmonary tuberculosis instead of only

**Data availability statement:** The data that support the findings of this study have been deposited in figshare at 10.6084/m9.figshare.28785116 (notification and contact tracing register) and 10.6084/m9.figshare.28785119 (TPT register).

**Funding:** The modular training under this SORT IT course was funded by the ICMR-National Institute of Epidemiology (ICMR-NIE), Chennai, India. ICMR-NIE did not provide any specific funding for the operational/ implementation research (that resulted in this manuscript) conducted through this SORT IT course. The research was conducted in routine operational settings utilizing existing health resources and workforce. Open access publication fee was supported by Mahatma Gandhi Institute of Medical Sciences, Sevagram, India and ICMR-National Institute of Epidemiology, Chennai, India.

**Competing interests:** The authors have declared that no competing interests exist.

microbiologically positive ones may improve treatment outcome recording among clinically diagnosed cases. The capacity building, monitoring, and supportive supervision need further strengthening to improve the provision of tuberculosis preventive treatment care.

## Introduction

The Sustainable Development Goals targets to control tuberculosis (TB) as a public health problem and envisages a reduction in the incidence rates by 80% of 2015 levels by the year 2030 [1]. Tuberculosis infection (TBI) is a non-infectious state when viable bacilli is present in a person along with host response but without any signs, symptoms or macroscopic pathology consistent with tuberculosis [2]. An estimated 2 billion people globally are infected with TB [3]. About 10% of those with TBI develop TB disease sometime in near future [4]. Household contacts (HHC) of people with TB are at higher risk of getting infected. Previous studies have estimated that around 14–20% of transmission of TBI could be attributed to household transmission [5–7]. According to systematic reviews from India [8] as well as globally [9], around 50% household contacts (HHC) of people with pulmonary TB (PwPTB) harbour TBI. The rate of progression from TBI to TB disease among such HHC was 12 per 1000 person-years [10].

To accelerate the rate of annual decline in TB incidence, the National Tuberculosis Elimination Programme (NTEP) is employing multiple strategies, one of which is Tuberculosis Preventive Treatment (TPT) [11]. Considering the fact that all HHC are at increased risk of TBI transmission, the NTEP, in the year 2021 expanded the scope of TPT services to HHC over 6 years in addition to the already existing strategy for children under 6 years [1,11].

As TPT among HHC over 6 years is a newer initiative, there is limited evidence of TBI and TPT management from routine programme settings. A recent study discusses the risk factors for TPT non completion from West Bengal, India [12]. Some studies discuss the losses in TPT care cascade among children [13,14]. To the best of our knowledge, there are no studies, describing the TPT care cascade in programme settings in India. However, studies from few districts of Maharashtra, Karnataka and Delhi, conducted in research settings discuss the care cascade and report challenges pertaining to the processes, logistics, training, community engagement and acceptance of TPT leading to losses in the care cascade and thus missed opportunities [15–17].

In India, the state of Maharashtra is one of the high TB burden states contributing to 12% of country's burden of PwPTB (0.15 million) in 2023, but the state could put around 48,000 HHC on TPT, which represents 6% of the HHC initiated on TPT in India [18]. This suggests losses during the TPT care cascade, which needs to be quantified at each step to understand the current TBI and TPT situation in Maharashtra. We believe, evidence of TPT care cascade from a large, high TB burden state will help developing strategies or interventions for TBI management and achieve the desired TB control in India.

We therefore planned this study with the objective of describing the TPT care cascade for HHC of all notified PwPTB for the year 2023 in the state of Maharashtra. Based on the findings of this study, we plan to develop and implement interventions in routine programme settings to strengthen the care cascade.

## Methods

### Study settings

**General setting.** Maharashtra is one of the bigger state in India with an area of about 3,00,000 square kilometers and a population of 130 million across 36 administrative districts divided administratively into 6 divisions [19]. The state in situated in western part of the country, bordering six other states and the Arabian sea on one side. The state comprises 45% urban population and 9% tribal population [20].

**Specific setting.** Maharashtra has 80 NTEP districts providing services through 528 TB units. The state follows the national guidelines, according to which, the contact tracing should be done within a week of diagnosis of the index PwPTB and HHC, i.e., a person who shared the same enclosed living space as the index PwPTB for one or more nights or for frequent or extended daytime periods during the three months before the start of current TB treatment.

As per the guidelines, all HHC are be initially screened for TB symptoms. Symptomatic HHC are evaluated for presence of active TB disease using chest X-ray and bacteriological examination of sputum, and only those free of active disease are considered for TBI evaluation. Among the asymptomatic contacts aged 6 years or more, active TB disease is ruled out using chest X-ray. TBI is to be evaluated using tuberculin sensitivity test or interferon gamma release assay as per availability. HHC with normal X-ray and positive TBI test results are considered eligible for TPT. HHC aged 5 years or less need not undergo TBI tests and are considered eligible for TPT if free of active TB disease.Due to reasons such as limited availability/ accessibility of tests, unwillingness of HHC, lack of skilled human resources etc. some HHC may not undergo the required investigations. HHC unable to undergo TBI testing due to any of these reasons are also considered eligible for TPT if they are free of active TB disease, consistent with national guidelines. A flowchart depicting the care cascade for TPT has been shown in Fig 1 [11,18].

In addition to 6H regimen (6 months of daily isoniazid), several new shorter regimens of TPT have emerged such as 1HP (one-month of daily Isoniazid and Rifapentine) for HIV positive HHC, 3HP and 3RH (three-months of weekly Isoniazid and Rifapentine, and three months of daily Isoniazid and Rifampicin) for HHC of drug sensitive TB, 4R and 6 Lfx (four-months of daily Rifampicin, and six-months of daily levofloxacin) for HHC of drug resistant TB are also available. During the year 2023, for HHC of drug sensitive TB, 6H was the majorly available regimen in Maharashtra with limited availability of 3HP and 3RH. TPT was provided to the beneficiaries as per the indications and availability [11,18].

Contact tracing is the primary responsibility of frontline workers viz. ASHA worker (Accredited Social Health Activist). The symptomatic screening of the enlisted contacts and referral for further assessment of eligibility is the primary responsibility of the Community Health Officer at the health and wellness centre – sub centre. Further assessment of the contacts for TBI and TB disease is done at the primary health care. The senior treatment supervisor is responsible for supportive supervision of ASHA workers and support to beneficiaries for treatment adherence [11].

The senior treatment supervisor uses *Ni-kshay* portal, a web-based patient management system for recording and reporting data related to TPT cascade for HHC in 3 electronic registers [21]. All index PwPTB are registered in TB notification register and identified by a unique patient id. Whether contact tracing was done for the index PwPTB is also recorded here. The contact tracing register records the aggregate number of contacts identified for each PwPTB, those screened for TB, evaluated for TB and TBI, eligible for TPT and numbers initiated on TPT. The TPT register contains the individual details of all persons who are initiated on TPT along with the dates and treatment outcomes.

### Study design

A retrospective cohort study was conducted based on routinely collected programmatic data extracted from the notification register, contact tracing register, and TPT register of the web-based patient management system of NTEP (*Ni-kshay*).

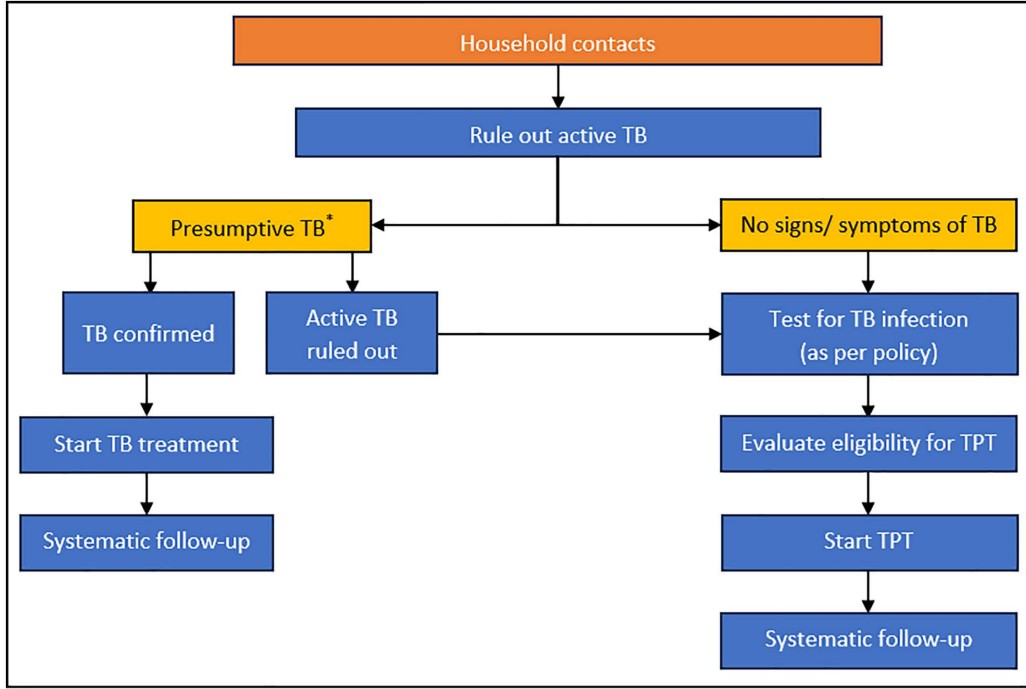

*Presumptive TB: Any one of cough or fever or night sweats or haemoptysis or weight loss or chest pain or shortness of breath or fatigue. In children <5 years, they should also be free of anorexia, failure to thrive, not eating well, decreased activity or playfulness to be considered asymptomatic.

**Fig 1. TPT care cascade as per the guidelines on programmatic management of tuberculosis preventive treatment in India.**

## Study population

The study population included 1) index PwPTB diagnosed and notified in Maharashtra between January and December 2023 and 2) household contacts of all PwPTB notified between January and December 2023.

## Data management and analysis

Data, extracted in MS excel format, from the three registers were imported in SPSS v20 software. Data from notification and contact tracing registers were merged using the unique patient id. People with extra pulmonary TB, site not recorded, those notified outside the specified period (Jan-Dec 2023), and outliers, i.e., the records with total number of HHC > 20 or number of HHC (aged ≤ 6 years) >10 were excluded. All the identifiers other than unique patient id were removed prior to performing the analysis. The data was downloaded in the last week of November 2024.

The outcomes were assigned for the HHC as per the national guidelines. For the purpose of this study, we operationally classified the treatment outcomes as TPT completed, TPT not completed and outcome not recorded. Outcomes labelled as died, lost to follow-up, treatment failed, TPT discontinuation due to toxicity, and not evaluated were categorized as 'TPT not completed'. The HHC without reports of TPT outcome were categorized as 'Outcome not recorded. Categorical variables were summarized as frequency and percentages. Continuous variables were summarized as median (Inter-quartile range, IQR). We also assessed the TPT outcomes by sociodemographic characteristics of the HHC, TPT regimen and clinical characteristics of index PwPTB.

### Ethics statement

Ethics approval was obtained from Institutional human ethics committee, National Institute of Epidemiology, Chennai (NIE/IHEC/A/202408–03 dated 18/09/2024) and Institutional ethics committee, Mahatma Gandhi Institute of Medical Sciences, Sevagram, Wardha (MGIMS/IEC/COMMED/187/2024 dated 31/8/2024) before the initiation of the study.

## Results

### Contact tracing and enlisting household contacts

A total of 223,827 TB patients were notified in 2023 in Maharashtra. After excluding people with extra pulmonary TB (n = 80,656), those with disease site not reported (n = 9944), those with enrolment dates outside 2023 (n = 32) and outliers with respect to number of HHC (n = 28), we included 133,167 PwPTB for analysis. Contact tracing was done for 112,034 (84.13%) PwPTB through which a total of 406,291 HHC were identified. Of the identified HHC 381,326 (94%) were aged 6 years or more. The median number of enlisted HHC per PwPTB was 3 (IQR: 2–4).

### Tuberculosis preventive treatment care cascade

Overall, 406,291 HHC were enlisted through contact tracing and of them 386,224 (95%) were screened for symptoms of TB (Fig 2). Among the screened contacts, 7,632 (2%) were symptomatic and 378,592 (98%) were asymptomatic. Of the symptomatic HHC, 6,348 (83%) were evaluated for presence of TB disease. Of the symptomatic contacts who were evaluated for presence of TB disease, 799 (13%) were diagnosed with TB whereas 5,549 (87%) did not have TB. 649 (81%) of those detected with TB were initiated on TB treatment. Of the total HHC, 185,502 (46%) were eligible for TPT among whom, 101,325 (55%) were initiated on TPT. This amounts to around 25% of the enlisted HHC. Of those who were started on TPT, 89,506 (88%) were initiated on the same day of the diagnosis of the index PwPTB.

Out of all HHC initiated on TPT, treatment outcomes were reported for 44,134 (44%) HHC and the outcomes for remaining 57,191 (56%) were not recorded. Among those for whom the TPT outcomes were reported, 41,480 (41%) HHC successfully completed TPT, and 2,654 (3%) HHC had not completed TPT. The characteristics of the HHC initiated on TPT and respective treatment outcomes are described in Table 1. The proportions of individual treatment outcomes for those categorized as TPT not completed is given in Fig 2. Six TPT regimens were administered with majority receiving 6H regimen (86,497; 85%) followed by 3HP regimen (5,364; 5%). TPT completion was similar for all age groups and all the genders. It was higher among those on shorter regimens, i.e., 3HP (78%), 4R (72%), and 3RH (56%) as compared to those on 6H regimen (41%). Treatment outcomes for 56% of the contacts was not recorded. Missing data was higher if the index patient sought care from private sector (63%), among contacts of clinically diagnosed cases (70%), and among those administered 6H regimen (57%) as compared to the other shorter regimens.

## Discussion

There were considerable losses in the TPT care cascade in Maharashtra, including contact tracing of index PwPTB and TPT completion of HHC. Around half of the eligible contacts were initiated on TPT. TPT outcomes for over half of the HHC put on treatment were not recorded in the *Ni-kshay* portal. The proportion of missing treatment outcomes was higher for those initiated on 6H regimen, private sector patients, and clinically diagnosed cases.

India, as a whole could put around 24% of the enlisted HHC on TPT [18] and globally this proportion is 21% [22], which are quite similar to our findings. Although this proportion varies from 15% - 100% across several settings throughout the world, results from pooled analysis reveal initiation of TPT in over 90% of the eligible HHC [23,24]. Fear of side effects [15,17], low risk perception among asymptomatic HHC, and lack of proper information and counselling [13,24,25] are important reasons for low acceptance of TPT. High workload of the staff also limits their ability to follow up and ensure that all eligible HHC are initiated on TPT [24,26].

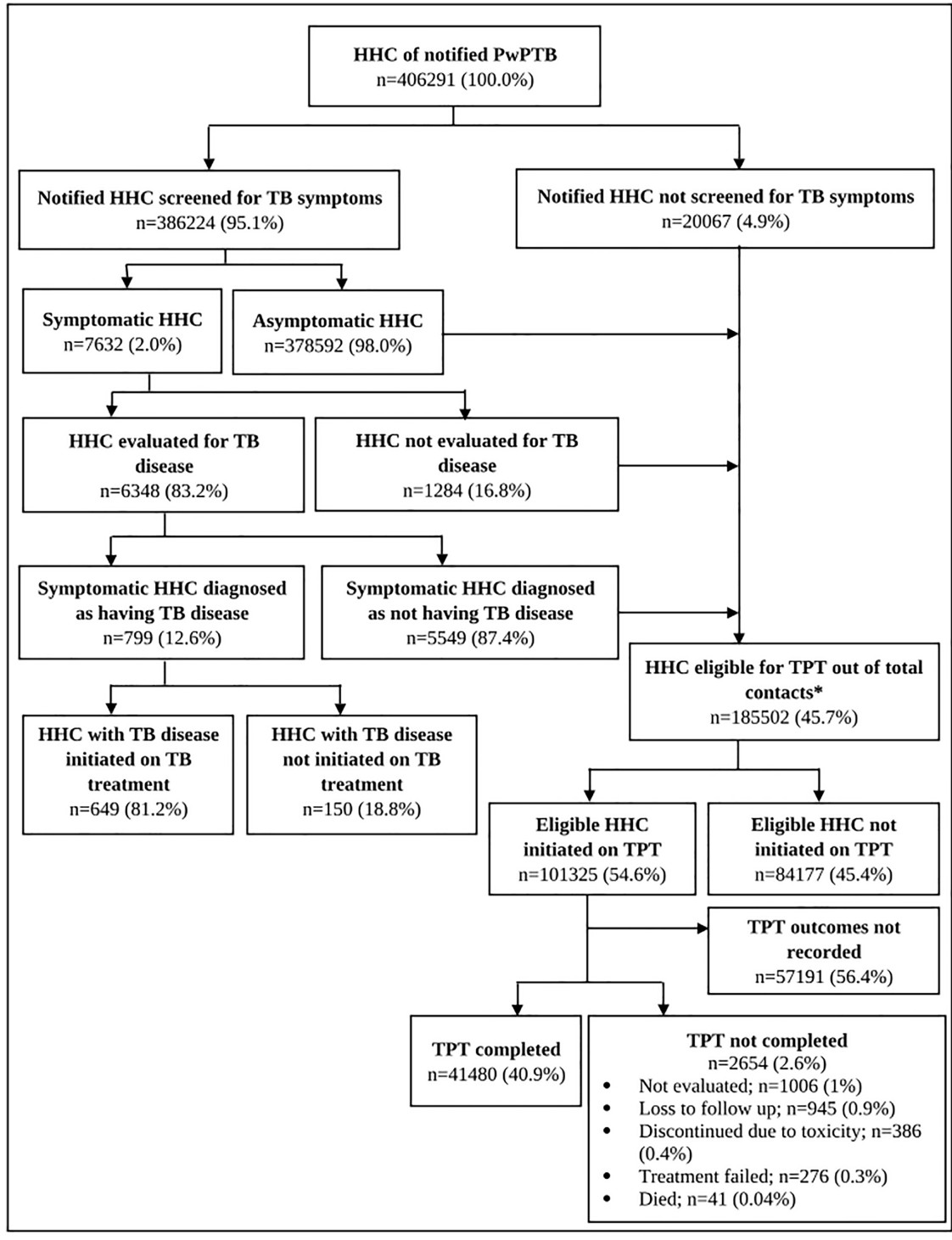

PwPTB: people with pulmonary tuberculosis, HHC: household contact, TPT: Tuberculosis Preventive Treatment.

\* As per data source (contact tracing register)

**Fig 2. Tuberculosis preventive treatment care cascade among household contacts of PwPTB in Maharashtra, India during 2023 (N = 406,291).**

**Table 1. Select characteristics of HHC of PwPTB by TPT treatment outcomes in Maharashtra, India during the year 2023.**

| Characteristics | TPT outcomes | | | |
|---|---|---|---|---|
| | Total N | Treatment completed n (%) # | TPT not completed$ n (%) # | Outcomes not recorded n (%) # |
| **Household contact (HHC) characteristics (N=1,01,325)** | | | | |
| **Overall** | 101,325 | 41,480 (40.9) | 2654 (2.6) | 57,191 (56.5) |
| Age groups | | | | |
| <6 years | 10,185 | 4,205 (41.3) | 235 (2.3) | 5,745 (56.4) |
| 6–17 years | 19,800 | 8,105 (40.9) | 475 (2.4) | 11,220 (56.7) |
| 18–44 years | 44,925 | 18,584 (41.4) | 1,186 (2.6) | 25,155 (56.0) |
| 45–59 years | 16,754 | 6,892 (41.1) | 459 (2.7) | 9,403 (56.1) |
| ≥60 years | 9,661 | 3,694 (38.2) | 299 (3.1) | 5,668 (58.7) |
| Gender | | | | |
| Male | 50,559 | 20,720 (41.0) | 1,318 (2.6) | 28,521 (56.4) |
| Female | 50,660 | 20,706 (40.9) | 1,331 (2.6) | 28,623 (56.5) |
| Transgender | 105 | 54 (51.5) | 5 (4.8) | 46 (43.8) |
| Current health facility type | | | | |
| Public | 95,503 | 39,291 (41.1) | 2,451 (2.6) | 53,761 (56.3) |
| Private | 5,822 | 2,189 (37.6) | 203 (3.5) | 3,430 (58.9) |
| Type of TPT regimen | | | | |
| 1 HP | 1 | 0 (0.0) | 0 (0.0) | 1 (100.0) |
| 3 HP | 5,364 | 4,162 (77.6) | 360 (6.7) | 842 (15.7) |
| 3 RH | 122 | 68 (55.7) | 18 (14.8) | 36 (29.5) |
| 4 R | 92 | 66 (71.7) | 3 (3.3) | 23 (25.0) |
| 6 H | 86,497 | 35,276 (40.8) | 1,698 (2.0) | 49,523 (57.3) |
| 6 Lfx | 313 | 131 (41.9) | 7 (2.2) | 175 (55.9) |
| Regimen not recorded | 8,936 | 1,777 (19.9) | 568 (6.4) | 6,591 (73.8) |
| **Characteristics of Index PwPTB (N=100,450)@** | | | | |
| Basis of diagnosis | | | | |
| Microbiologically confirmed | 87,681 | 37,438 (42.7) | 2,295 (2.6) | 47,948 (54.7) |
| Clinical diagnosis | 12,769 | 3,608 (28.3) | 265 (2.1) | 8,896 (69.7) |
| Health sector of index patient | | | | |
| Public | 88,218 | 36,877 (41.8) | 2,207 (2.5) | 49,134 (55.7) |
| Private | 12,232 | 4,169 (34.1) | 353 (2.9) | 7,710 (63.0) |

*HHC – household contact, TPT – tuberculosis preventive treatment, PwPTB – people with pulmonary tuberculosis.

#All percentages in parenthesis are row percentages.

$TPT not completed – this has been operationally defined for this study as all treatment outcomes other than treatment completed.

@information on basis of diagnosis and health sector of index patient not available for 875 patients.

The state of Maharashtra enlists a median 3 contacts per PwPTB. The mean contacts enlisted per PwPTB from other Indian studies ranged from 2.9 – 3.4 [15–17]. Contact tracing and identification of all HHC is the first step of the cascade. As over half of the HHC of PwPTB have TBI [8], the identification of all HHC becomes very critical. High workload [17], unavailability of HHC during home visit [15,24], and stigma are some of the reasons of losses at this step of the cascade.

Identification of HHC is followed by their evaluation for TB disease and TBI. TPT provision to HHC with TBI, reduces their hazard of developing the disease by over 80%, and thus, proper and efficient execution of this step is important to ensure effectiveness of TPT [26,27]. Alsdurf et al. [23] found that only around two-thirds of those intended undergo the

required evaluation. These evaluations, including those to rule out TB, mostly demand travel of HHC to health facilities adding to the time and money that one has to spend, thereby aggravating the challenge. TB programmes need to devise strategies to bring these services closer to the doorsteps of the beneficiaries along with greater community involvement for better utilization [28]. Issues like limited availability of TBI testing services, unavailability/ unwillingness of identified HHC for tests, lack of awareness [15], and stigma [16] further complicate the challenges. Many national TB programmes are providing TPT to all HHC free of TB when TBI testing cannot be done as is the case in India. This strategy may reduce the loss of HHC with TBI, but also increase the associated costs as many HHC initiated on TPT might have turned negative results for TBI and hence would have been ineligible for TPT. Studies argue that this approach also increases the risk of harm from unnecessary TPT consumption. In case active TB is not ruled out before initiating HHC on TPT, especially among those with subclinical TB, it will result in suboptimal treatment for such people [29,30].

In our study the reported TPT completion was low. Missing data about treatment outcomes, the pattern of which we are not sure about, limits our ability to comment on over or under reporting of non-completion of TPT. 6H was the most commonly available regimen at the time of this study with 3HP and 3RH regimens being available in limited settings. The programme is gradually shifting on 3HP regimen for adult HHC, which as per available evidence has better TPT completion and similar safety profile [31–33]. Studies from India [17] and Peru [34], suggest that beneficiaries prefer shorter regimen as the dosage schedule are easy to remember and less disruptive. TB programme workforce is accustomed to monitor treatment outcomes among the patients and hence, monitoring and follow up of apparently healthy contacts, as in the case of TPT may be challenging specially with longer treatment regimens [26]. This could possibly explain the higher rate of missing treatment outcomes in our study especially among those on longer TPT regimen.

In our study, the recording of outcomes as well as TPT completion was lesser among HHC of clinically diagnosed cases than that among HHC of microbiologically confirmed ones. The risk of TPT non-completion among HHC of clinically diagnosed PwPTB is 1.6 times as compared to HHC of microbiologically confirmed PwPTB [12]. The national guidelines recommend initiation of TPT among HHC of all PwPTB, however, the India TB report does not show TPT indicators for HHC of clinically confirmed PwPTB [18]. The India TB report needs to be aligned to the guidelines to minimize any complacency that may result due to such differences. The recording of outcomes was lower among patients and/or contacts seeking care from private sector. Such contacts are also at higher risk of TPT non-completion and need better monitoring [12]. There are reports of hesitancy in the private sector regarding adapting TPT services [35]. This might have affected the treatment outcomes among those seeking care from private sector in our study.

Our study had few limitations, mainly due to the fact that the study was based on analysis of secondary data available with the national programme. First, we were unable to analyze the delays in the cascade and time to events due to missing outcomes and their dates. Second, the data from contact tracing register was an aggregate one due to which, we were unable to understand the proportions of losses from the cascade following symptomatic screening for TB and during TBI evaluation. Although, the contact tracing register captures numbers evaluated for and diagnosed with TB disease, it does not capture the number of HHC evaluated for TBI or those diagnosed with TBI, hence, we are unable to assess the gaps in TBI evaluation of HHC under the national programme. Third, the possibility of underreporting of undesirable outcomes among those for whom the outcomes were not reported cannot be ruled out.

## Recommendations

The aspect of TPT provision for HHC ≥ 6 years, is a newer addition in the programme. From this statewide data analysis, we identified losses in the TPT care cascade and incomplete recording of information, particularly treatment outcomes. Capacity building of the health care staff including those from private sector, responsible for implementing TPT, routine review of TPT indicators and data completion, strengthening the monitoring and supportive supervision and further improvement in data capturing mechanisms is required. The programme also needs to identify the reasons for losses from the cascade and address them. Since the quantum of TPT beneficiaries is large, decentralized mechanisms for

identification, evaluation and TPT provision are necessary for improvement in services delivery and utilization. We recommend that annual India TB report should include TPT related indicators for HHC of all PwPTB (microbiologically confirmed and clinically diagnosed) instead of only microbiologically confirmed PwPTB, thereby aligning the indicators in annual reports with the national guidelines.

## Supporting information

**S1 File. This is the strobe checklist.**
(DOCX)

## Acknowledgments

We would like to thank Dr Ashok Randive and Mr Waghmare from the state NTEP office, Mr Sumant Dhoble and Mr Punvatkar from District TB centre, Wardha for their help in data acquisition and inputs during the process of data cleaning and analysis.

This operational/ implementation research that resulted in this manuscript was conducted through the Structured Operational Research and Training Initiative (SORT IT), a global partnership led by the Special Program for Research and Training in Tropical Diseases at the World Health Organization (WHO/TDR). The model is based on a course developed jointly by the International Union Against Tuberculosis and Lung Disease (The Union) and Medécins sans Frontières (MSF/ Doctors Without Borders). This specific SORT IT course which resulted in this publication was part of year one of ICMR-National Institute of Epidemiology (ICMR-NIE) led TB SORT IT course 2024–26, with support and guidance from India's Central TB Division and WHO India. It was jointly developed and implemented by: ICMR-National Institute of Epidemiology (ICMR-NIE), Chennai, India; ICMR-National Institute for Research in Tuberculosis (ICMR-NIRT), Chennai, India; Post Graduate Institute of Medical Education and Research (PGIMER), Chandigarh, India; FIND, New Delhi, India; Baroda Medical College, Vadodara, India; Narotam Sekhsaria Foundation, Mumbai, India; Government Medical College, Shahdol, India; All India Institute of Medical Sciences (AIIMS), Madurai, India; All India Institute of Medical Sciences (AIIMS), Bhathinda, India; Yenepoya Medical College, Mangaluru, India; and GMERS Gotri Medical College, Vadodara, India.

## Author contributions

**Conceptualization:** Anuj Mundra, Tarun Bhatnagar, Mrinalini Das, Aniruddha Kadu.

**Data curation:** Anuj Mundra, Sandeep Sangale, Abhishek Raut.

**Formal analysis:** Anuj Mundra.

**Investigation:** Anuj Mundra.

**Methodology:** Anuj Mundra, Tarun Bhatnagar, Mrinalini Das.

**Project administration:** Anuj Mundra.

**Resources:** Sandeep Sangale, Bishan S Garg.

**Software:** Anuj Mundra, Sandeep Sangale, Hemant Patil.

**Supervision:** Tarun Bhatnagar, Mrinalini Das, Rameshwar J Paradkar, Subodh S Gupta, Bishan S Garg.

**Validation:** Aniruddha Kadu.

**Visualization:** Anuj Mundra, Tarun Bhatnagar, Mrinalini Das.

**Writing – original draft:** Anuj Mundra.

**Writing – review & editing:** Tarun Bhatnagar, Mrinalini Das, Sandeep Sangale, Hemant Patil, Abhishek Raut, Aniruddha Kadu, Rameshwar J Paradkar, Subodh S Gupta, Bishan S Garg.

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
