## [Decision Letter · Decision Letter 0]

12 May 2025

PGPH-D-25-00926

Identifying missed opportunities in tuberculosis preventive treatment care cascade: Analysis of programme data from Maharashtra, India

Dear Dr. Mundra,

Thank you for submitting your manuscript to PLOS Global Public Health. After careful consideration, we feel that it has merit but does not fully meet PLOS Global Public Health’s publication criteria as it currently stands. Therefore, we invite you to submit a revised version of the manuscript that addresses the points raised during the review process.

We look forward to receiving your revised manuscript.

Kind regards,

Anete Trajman

Academic Editor

Journal Requirements:

Additional Editor Comments:

Although the findings are not novel, we feel that there is potential for publication if the issues raised by the reviewers and by myself are addressed.

Clearly, the national (or state) guidelines need to be clarified for the audience to understand your results, as pointed out by the reviewers. Maybe a figure on the algorithm for HHC management would help the reader to understand your numbers. It is not clear why so many people were considered not eligible, and sentences such as lines 173-174 (41% completed and 3% did not complete) are not clear. In this same setting subheading, please explain what regimens are available and how are they selected. Healthcare worker or patient's choice? Are there specific indications for different regimens? If so, this might impact on treatment outcomes and should be considered in the interpretation of findings.

Please justify your choices of outcomes for comparison against completed, which are so different and have very different meanings. A complementary analysis of completed of TPT completion versus biological outcomes could bring light to your findings and discussion.

Your discussion needs thorough revision. It should help the reader to understand the meaning of your findings. Do not repeat (twice) the results in the discussion, just summarize them in the first paragraph and then interpret them under the light of the literature and of the limitations of the study. Please incude the very high prevalence of TB found in contacts, this has very relevant public health implications on recommendations for contact tracing as well. Think on what could explain your results. In this matter, please cite the possible bias of non-completion results (or other outcomes) among those who were not reported. Also mention the nature of this secondary data observational study (the design) as a limitation. Your conclusions, except form the first sentence, are not conclusions from your data. Please change this subheading to Recommendations or Final considerations.

A STROBE check list would be helpful, please attach one in your resubmission.

Minor points:

Please update your definition of TBI (Coussens et al) in the Introduction. Please update your references and rather use systematic reviews instead of single studies. Global reports are more reliable than European data to discuss your findings.

When revising, please amend the numbers to the occidental annotation. We understand the authors used the Indian numeric system, but this is an international journal, read by a global audience.

Reviewers' comments:

Reviewer's Responses to Questions

**Comments to the Author**

1. Does this manuscript meet PLOS Global Public Health’s publication criteria?

Reviewer #1: Yes

Reviewer #2: Partly

2. Has the statistical analysis been performed appropriately and rigorously?

Reviewer #1: Yes

Reviewer #2: No

3. Have the authors made all data underlying the findings in their manuscript fully available (please refer to the Data Availability Statement at the start of the manuscript PDF file)?

Reviewer #1: Yes

Reviewer #2: Yes

4. Is the manuscript presented in an intelligible fashion and written in standard English?

Reviewer #1: Yes

Reviewer #2: No

Reviewer #1: Paper of interest to public health, but needs to be clearer in some items, namely:

Line 110: The authors refer to other TPT regimens, but do not describe them. These regimens will only appear in the results.

Line 115: PHC: this acronym needs to be translated

Line 141: Why did the authors consider as complete treatment those who died, were lost to follow-up, interrupted treatment due to toxicity and those without a final evaluation? I suggest excluding them from this category, showing the percentage found.

Finally, in the Methodology and Results sections, I suggest including a figure including the “n” of each stage. The numbers presented are very confusing, causing the reader to have to go back to the original numbers to understand the cascade presented.

Reviewer #2: Thank you for the opportunity to review the manuscript titled “Identifying missed opportunities in tuberculosis preventive treatment care cascade: analysis of programme data from Maharashtra, India.”

Abstract:

-Use of commas in the reported numbers are inconsistent in use and placement.

-Authors state that “household contacts of all people with pulmonary tuberculosis after ruling out active disease.” You then note that only 45% of HHCs were listed as eligible for TPT. Is that because others had active disease? Or because it was only prescribed in 45%? Would be helpful to understand who was diagnosed with active disease to remove from the cascade for the next step.

Introduction:

-“ One recent strategy of NTEP is focusing on screening of TB and TPT provision in HHC (aged more than 6 years) of PwPTB..” What about children 5 and under? They are typically a group who are vulnerable and recommended TPT when exposed.

-It is noted that there are existing studies reporting challenges and losses in the care cascade already in districts of Maharashtra. How is this study different—includes the entire state of Maharashtra? Is it updated and including a newer time period? Were lessons learned from the previously identified challenges—processes, logistics, community engagement, acceptance—related to losses in the care cascade addressed since the last study? If so, should we expect to see improvement across the cascade? Please provide more clear rationale for why this study was conducted.

Methods:

-Study setting and specific TB epidemiology seem spread across the introduction and methods; would be better to consolidate.

-It sounds like current strategy is to rule out TB disease, test (TST or IGRA) for TBI, and then provide TPT. However, if testing is not available, they are also provided with TPT. Can you please clarify this—is there a time period at which it is determined they need TPT despite no test available. Are there certain facilities that do not offer TST or IGRA? What is the variability of procedures/testing availability across the study area, as this will inform eligibility for TPT determination and potential delayed initiation?

-What are procedures/eligibility to determine who gets which regimen? Regimen type may differentially impact completion rates based on duration, side effect, etc.

-The description of who is responsible for what (lines 111-117) are unclear. Also, what type of capacity building and supportive supervision and support for treatment adherence are provided? Are these actually adhered to?

-When contact tracing is done, is it just household contact or close contacts? Please provide the definition.

-Did the index patient need to have bacteriologic confirmation of disease in order for contact tracing to be done?

-Please clarify who was included. Did the diagnosis of pulmonary TB for the index patient need to take place from Jan-Dec 2023? If so, it takes time for contact tracing, testing, screening, treatment initiation, and completion. Were contacts followed for a longer period of time to make sure they had time to complete?

-Does ‘not evaluated’ indicate a response was missing or that not enough time had passed and they could still be on treatment at the time of data download?

Results:

-Same comment as earlier about commas in numeric values.

-Please clarify the exclusionary criteria of ‘outliers with respect to number of HHC’. How was this defined? Please include in methods.

-Why were only 1,85,502 HHC eligible for TPT? Did others have active disease? Shouldn’t everyone except those with active disease be eligible?

-It seems like Line 161 should come after Lines 162-166.

-If only 3% of HHCs did not complete TPT, what were the reasons, and what happened to the rest if only 41% successfully completed? Later it says that treatment outcomes for 56% of the contacts was not recorded. Is this because there was not time to complete it before data were recorded or collected? I think this is a major issue that is not addressed or defined appropriately in the manuscript.

-Results talk about stratification by private vs public sector, but there was no information about how services or the patients who attend these facilities differ.

**Do you want your identity to be public for this peer review?** For information about this choice, including consent withdrawal, please see our Privacy Policy

Reviewer #1: No

Reviewer #2: No

---

## [Decision Letter · Decision Letter 1]

30 Jul 2025

PGPH-D-25-00926R1

Identifying missed opportunities in tuberculosis preventive treatment care cascade: Analysis of programme data from Maharashtra, India

Dear Dr. Mundra,

Thank you for submitting your manuscript to PLOS Global Public Health. After careful consideration, we feel that it has merit but does not fully meet PLOS Global Public Health’s publication criteria as it currently stands. Therefore, we invite you to submit a revised version of the manuscript that addresses the points raised during the review process.

We look forward to receiving your revised manuscript.

Kind regards,

Anete Trajman

Academic Editor

Journal Requirements:

Additional Editor Comments (if provided):

The authors have improved their manuscript but major clarifications are necessary. The methods are still unclear, especially the eligibility criteria. This makes the results look unsound. We are giving the authors one additioal opportunity to make the eligibility criteria and the findings clear before a final decision.

Reviewers' comments:

Reviewer's Responses to Questions

**Comments to the Author**

Reviewer #1: (No Response)

Reviewer #3: (No Response)

publication criteria?

Reviewer #1: Partly

Reviewer #3: (No Response)

3. Has the statistical analysis been performed appropriately and rigorously?

Reviewer #1: Yes

Reviewer #3: (No Response)

4. Have the authors made all data underlying the findings in their manuscript fully available (please refer to the Data Availability Statement at the start of the manuscript PDF file)?

Reviewer #1: Yes

Reviewer #3: (No Response)

5. Is the manuscript presented in an intelligible fashion and written in standard English?

Reviewer #1: Yes

Reviewer #3: (No Response)

Reviewer #1: The authors have significantly improved the paper, but clarifications are still needed, namely:

Methodology:

Line 129 – Those who did not receive skin testing or other tests: Why? Was there no availability at the time of screening? I suggest adding clarification.

The authors should be clearer about how HHC were extracted from the reporting system, as well as clarify how the TPT regimen were chosen for HHC (only availability at the time of treatment initiation or was there another criterion?)

Figure 1:

• Needs a title.

• Does the target population refer to contacts? This needs to be made clear.

• What do the authors call presumptive TB? Have these HHC presented some symptoms?

• Evaluate for TPT means eligibility criteria?

Results:

Figure 2:

• Needs a title

• The numbers in Figure 2 were not changed, so they need to be corrected to align with the text.

Discussion:

Line 266 – How were contacts diagnosed with active TB (X-ray? Physical examination? Bacteriological examination?)

Reviewer #3: Thank you for the opportunity to review the revised manuscript, which provides valuable insights into the TPT care cascade among household contacts of pulmonary TB cases in Maharashtra, India. The manuscript addresses an important public health topic and is generally clear and well-structured. However, I would like to highlight specific areas where additional clarification would significantly improve the manuscript.

One important issue requiring more attention is the criteria used to determine eligibility for TPT, especially regarding contacts who were not adequately assessed for symptoms or whose symptom assessment status was unclear. Currently, it is not entirely clear how contacts who were either symptomatic and not thoroughly evaluated or asymptomatic and did not have confirmatory tests (IGRA or TST) were classified as eligible for TPT.

Specific suggestions for enhancing clarity:

1. Methods Section:

It would be helpful to include a brief but explicit description of the criteria used for TPT eligibility, particularly for contacts unable to complete TBI testing. For instance, consider specifying:

“Household contacts unable to undergo IGRA or TST testing were considered eligible for TPT if they were asymptomatic upon initial clinical screening performed by trained health professionals, consistent with national program guidelines.”

2. Results Section:

Indicate the number or proportion of contacts considered eligible based solely on symptom screening, without confirmatory IGRA or TST tests. For example:

“Among eligible contacts, XX% were included based exclusively on clinical screening (absence of symptoms), without IGRA or TST testing.”

3. Discussion Section:

Discuss the implications of determining TPT eligibility without confirmatory diagnostic testing, emphasizing potential limitations such as overtreatment or the risk of missing subclinical active TB cases.

Additionally, the results would benefit from explicitly reporting percentages or numbers at each stage of the diagnostic process.

**Do you want your identity to be public for this peer review?** For information about this choice, including consent withdrawal, please see our Privacy Policy

Reviewer #1: No

Reviewer #3: No

---

## [Decision Letter · Decision Letter 2]

22 Oct 2025

PGPH-D-25-00926R2

Identifying missed opportunities in tuberculosis preventive treatment care cascade: Analysis of programme data from Maharashtra, India

Dear Dr. Mundra,

Thank you for continuing to revise your submission. There are a few minor revisions still outstanding. Therefore, we invite you to submit a revised version of the manuscript that addresses the points raised during the review process.

We look forward to receiving your revised manuscript.

Kind regards,

Graeme Hoddinott, Ph.D

Academic Editor

Journal Requirements:

Additional Editor Comments (if provided):

Please address the minor comments from reviewer 1. Reviewer 4 also raised some interesting suggestions that I tend to agree with. If you are happy to accommodate these, then please do so.

Reviewers' comments:

Reviewer's Responses to Questions

**Comments to the Author**

Reviewer #1: (No Response)

Reviewer #4: All comments have been addressed

publication criteria?

Reviewer #1: Partly

Reviewer #4: Yes

3. Has the statistical analysis been performed appropriately and rigorously?

Reviewer #1: Yes

Reviewer #4: N/A

4. Have the authors made all data underlying the findings in their manuscript fully available (please refer to the Data Availability Statement at the start of the manuscript PDF file)?

Reviewer #1: Yes

Reviewer #4: Yes

5. Is the manuscript presented in an intelligible fashion and written in standard English?

Reviewer #1: Yes

Reviewer #4: Yes

Reviewer #1: The authors have adequately addressed the suggestions, but some corrections are still needed, such as:

English review for minor corrections

Line 184 - 84% of 133,167 is 111,860, not 112,034 (contact tracing)

Line 207 - TPT completion was similar for all age groups and for both the genders. I suggest “all genders” because the authors list three genders in the table

Reviewer #4: The manuscript covers important operational research identifying the losses across the Tuberculosis Preventive Treatment care cascade for household contacts of people with pulmonary TB in Maharashtra, India. It has a good flow and is written in an interesting way and is scientifically sound. I have a few suggestions to consider.

Title: It is accurate for the work done, although I have some reservation with the word ‘analysis’, as discussed below.

Objective: Simple and achivable and is purely descriptive.

Methods

Study Design: The authors have described study design as retrospective cohort study, as well as used the word ‘Analysis of programme data’ in their title. But it actually a descriptive record based study. In the standard epidemiological study classification, cohort study is classified under analytical studies and done to testing of hypothesis with the use of some effect size to measure the significance association. Like OR, RR, and performing parametric and or non paramatric tests. The present manuscript have not performed any testing of hypothesis, but only description of routinely collected data. It seem to fit more as described record beased study than a retrospective cohort study.

Study settings: Since Maharashtra is a large state and TPT is being delivered utilizing various mechanisms - Treat only, Test and treat, Montox, CyTB, public–private partnership, along with as a part of routine service. This need to be well documented and even observed across TPT treatment outcome in the table 1 if data is available.

Results: Well presented. Can we add OR in the tables?

**Do you want your identity to be public for this peer review?** For information about this choice, including consent withdrawal, please see our Privacy Policy

Reviewer #1: No

Reviewer #4: No

---

## [Decision Letter · Decision Letter 3]

19 Nov 2025

Identifying missed opportunities in tuberculosis preventive treatment care cascade: Analysis of programme data from Maharashtra, India

PGPH-D-25-00926R3

Dear Dr Mundra,

We are pleased to inform you that your manuscript 'Identifying missed opportunities in tuberculosis preventive treatment care cascade: Analysis of programme data from Maharashtra, India' has been provisionally accepted for publication in PLOS Global Public Health.

Best regards,

Graeme Hoddinott, Ph.D

Academic Editor

Reviewer Comments (if any, and for reference):

Reviewer's Responses to Questions

**Comments to the Author**

Reviewer #1: All comments have been addressed

Reviewer #4: All comments have been addressed

publication criteria?

Reviewer #1: Yes

Reviewer #4: Yes

3. Has the statistical analysis been performed appropriately and rigorously?

Reviewer #1: Yes

Reviewer #4: N/A

4. Have the authors made all data underlying the findings in their manuscript fully available (please refer to the Data Availability Statement at the start of the manuscript PDF file)?

Reviewer #1: Yes

Reviewer #4: Yes

5. Is the manuscript presented in an intelligible fashion and written in standard English?

Reviewer #1: Yes

Reviewer #4: Yes

Reviewer #1: The authors addressed all the comments and answered them satisfactorily. Therefore, I have no further comments.

Reviewer #4: All the comments have been adequately addressed.

**Do you want your identity to be public for this peer review?** For information about this choice, including consent withdrawal, please see our Privacy Policy

Reviewer #1: **Yes: ** LILIAN DE MELLO LAURIA

Reviewer #4: No
